# Within-host bayesian joint modeling of longitudinal and time-to-event data of *Leishmania* infection

**Felix M. Pabon-Rodriguez**[1,¤,*], **Grant D. Brown**[1], **Breanna M. Scorza**[2,3,‡], **Christine A. Petersen**[2,3,‡]

**1** Department of Biostatistics, The University of Iowa College of Public Health, Iowa City, Iowa, United States of America, **2** Department of Epidemiology, The University of Iowa College of Public Health, Iowa City, Iowa, United States of America, **3** Center for Emerging Infectious Diseases, The University of Iowa College of Public Health, Iowa City, Iowa, United States of America

☯ These authors contributed equally to this work.
¤ Current address: Department of Biostatistics and Health Data Science, Indiana University School of Medicine, Indianapolis, Indiana, United States of America
‡ BMS and CAP also contributed equally to this work.
\* fpabonrodriguez@gmail.com

**Data Availability Statement:** The authors confirm that the data analyzed supporting the findings of this study are available within the article and its supplementary materials.

## Abstract

The host immune system plays a significant role in managing and clearing pathogen material during an infection, but this complex process presents numerous challenges from a modeling perspective. There are many mathematical and statistical models for these kinds of processes that take into account a wide range of events that happen within the host. In this work, we present a Bayesian joint model of longitudinal and time-to-event data of *Leishmania* infection that considers the interplay between key drivers of the disease process: pathogen load, antibody level, and disease. The longitudinal model also considers approximate inflammatory and regulatory immune factors. In addition to measuring antibody levels produced by the immune system, we adapt data from CD4+ and CD8+ T cell proliferation, and expression of interleukin 10, interferon-gamma, and programmed cell death 1 as inflammatory or regulatory factors mediating the disease process. The model is developed using data collected from a cohort of dogs naturally exposed to *Leishmania infantum*. The cohort was chosen to start with healthy infected animals, and this is the majority of the data. The model also characterizes the relationship features of the longitudinal outcomes and time-to-death due to progressive *Leishmania* infection. In addition to describing the mechanisms causing disease progression and impacting the risk of death, we also present the model's ability to predict individual trajectories of Canine Leishmaniosis (CanL) progression. The within-host model structure we present here provides a way forward to address vital research questions regarding the understanding of the progression of complex chronic diseases such as Visceral Leishmaniasis, a parasitic disease causing significant morbidity worldwide.

**Funding:** Research reported in this article was supported by the National Institutes of Allergy and Infectious Diseases (NIAID) of the National Institutes of Health (NIH) of the United States of America under award number R01AI139267-03, and by the Masters of Foxhounds Association Foundation under award MFHA18441000. This work was also performed while BMS was supported by NIH/NIAID T32AI007260, which was the basis for data collection. The funders had no role in study design, data collection and analysis, decision to publish, or preparation of the manuscript. There was no additional external funding received for this study.

**Competing interests:** "The authors have declared that no competing interests exist.

**Abbreviations:** *PPT, Posterior Predictive Trajectories.

## 1 Author summary

The immune system is complex and its effectiveness against infection depends on a variety of host and pathogen factors. Despite numerous studies of *Leishmania* parasite infections, researchers are still discovering new connections between immune system components with hopes of better understanding how the immune system functions during *Leishmania* infection.

The development of tools for understanding, preventing, and predicting *Leishmania* infection outcomes is the main goal of this work. We present a computational model made using field-collected data during canine *Leishmania* infections. The model considers the interplay between three main components: parasite load, antibody level, and disease severity. The model explores how key inflammatory and regulatory elements of the immune response affect these main components, including T cell proliferation and important cytokine expressions such as protective interferon-gamma (IFN-$\gamma$) or inhibitory interleukin 10 (IL-10) [1]. Although the induction of CD4+ T helper 1 cell responses is considered essential for immunity against *Leishmania*, B cells and the production of *Leishmania*-specific antibodies have also been proposed to play an important role in disease progression [2]. In a simpler model, Pabon-Rodriguez et al. [3] showed antibody levels are dependent on pathogen load and canine Leishmaniasis (CanL) disease presentation. These high levels of *Leishmania* specific antibodies are observed in subjects with visceral Leishmaniasis (VL) and other severe forms of Leishmanial disease, and there is accumulating evidence that B cells and antibodies correlate with pathology [4]. In Section 2, we introduce Canine Leishmaniasis and discuss the importance of host-pathogen interaction with the immune response. Next, in Section 3, we introduce the data collection study, the variables utilized in this model, and define the clinical signs of *Leishmania* infection. In addition, this section explains how the presented model was constructed based on different techniques. A summary of model parameters, model implementation details, convergence diagnostics, and sensitivity analysis are also included. In Section 4, we provide summary results of how different model variables interact with one another and disease progression forecasts. In Section 5, we discuss the results and provide further recommendations and considerations.

## 2 Introduction

Infectious disease modeling has rapidly increased over the years and is utilized by academics and the public health community to study disease progression, estimate different epidemiological measures, and study the effects of treatments or interventions. Most modeling approaches focus on the spread of disease by using population dynamic models, where subjects are compartmentalized into several possible states such as being susceptible, exposed, infected, or recovered from a particular disease. These models often ignore randomness and variability and are instead deterministic. They typically offer a single estimate of the relevant parameters without considering the level of uncertainty. Given the appropriate data, our suggested statistical modeling technique is extremely flexible and may be modified to answer a wide range of hypotheses. In addition to between-subject variability found in healthy, immunized, or infected patient data, we also include within-patient variability in modeling parameters of pathogen load, inflammation, and regulation within the immune responses to pathogens over time. These characteristics of statistical techniques offer defined advantages over mathematical models. By combining multiple immunological parameters into a single hierarchical model, a Bayesian statistical model enables the modeling of increasingly complicated processes by using knowledge from previous studies and analyses. Each sub-level is a conditional-probability

model, which can be clear and simple even when the whole model reflects a complex, multi-layered phenomenon, like the immune response to a pathogen.

Here we build a Bayesian modeling strategy combining properties of longitudinal, time series, and survival models to investigate the effects of important specific immune response variables on the ability to accurately predict disease progression and identify responses associated with the risk of death due to CanL. Partitioning the entire infection process into different components enables users to obtain intuitive estimations of model quantities. The interpretations utilized in simpler statistical models still hold true, especially when utilizing regression formulations. This enables us to estimate the direct and indirect effects of covariates of interest and serves as a useful reference for interface design when the models are put into reality. Such models will enable us to evaluate the probability of an inflammatory response to immunization as well as the overall impact of treatment on long-term pathogen control.

## 2.1 Leishmaniasis

*Leishmaniasis* is a vector-borne disease caused by infection with protozoan parasites of the genus *Leishmania*. Dogs serve as the primary reservoir for human infection with *L. infantum* and the immunopathology of CanL is very similar to human disease [5]. Both human VL and CanL are chronic, progressive diseases that lead to a state of immunosuppression and symptomatic cases that are fatal without treatment [5]. Available treatments are not ideal and no vaccine exists for humans. There is an ongoing critical need for novel therapeutics for VL/CanL, however, the complicated nature of the immune response to *Leishmania* parasites has impeded progress.

Dogs infected with *L. infantum* parasites can maintain a state with no clinical signs for an extended period of time before progressing to a state with clinical manifestations of CanL; some may never experience clinical disease. There has been much research done to identify immune factors involved in maintaining subclinical infection with interesting results. There is a need to incorporate these findings with statistical modeling to establish a deeper and more comprehensive understanding of the relationship between specific immune response components and pathogen control. A Type 1 immune response with the secretion of IFN-$\gamma$, produced predominately by CD4+ T cells, is required for clinical protection to *Leishmania* parasites [6]. Solano-Gallego et. al. [7] assayed cytokine production by cells from dogs infected with *L. infantum* and found a significant negative relationship between IFN-$\gamma$ production and parasitemia. IFN-$\gamma$ is a highly inflammatory cytokine that suppresses parasite replication by activating the microbicidal activity of infected host cells to maintain a low pathogen load. Regulatory immune factors that antagonize the effects of IFN-$\gamma$ facilitate parasite replication. The immunoregulatory cytokine IL-10 is a potent antagonist of IFN-$\gamma$ [8–10]. Additionally, although also a marker of activation, regulatory surface receptors on T cells, such as programmed cell death 1 (PD-1), send inhibitory signals upon ligation to curb T cell effector functions such as proliferation, cytokine production, and cytotoxicity [11]. Esch et al. [1] investigated T cell PD-1 expression in dogs with progressive CanL and showed a role for this receptor in limiting the ability of T cells to induce parasite control within infected host cells. While CD8+ T cells are thought to play a relatively minor protective role in the immunopathogenesis of VL and CanL compared to their CD4+ counterparts, these cells express IFN-$\gamma$ and IL-10, to a lesser extent, and are well characterized in their expression of PD-1 [12, 13]. Finally, B cells produce antibodies during CanL, however, these antibodies are non-productive, and mainly correlate with an enhanced disease rather than limiting parasite burden [14]. These complex inflammatory and regulatory arms of the immune response exist in a delicate balance to eliminate pathogens while also limiting immune-mediated tissue pathology associated with sustained exposure to

inflammation. Understanding how different immune response components interact to impact the overall disease state would help identify candidates with immunotherapeutic potential.

Further complicating the situation, dogs with CanL experience a high rate of co-infections [15]. Pathogen composition can help model co-infection with different organisms or different strains of the same pathogen [16]. It is known that multiple pathways of the adaptive immune system coordinate to enact responses linked to different infectious agents [17]. By including both CD4+ and CD8+ T cell variables such as proliferation and expression of IFN-$\gamma$, IL-10, or PD-1 collected from a prospective cohort of dogs with CanL, we are able to more closely model real-world disease progression. In addition to these immune responses, the *Leishmania*-specific antibody level, pathogen burden, and disease state classification were also measured and considered within the model specification for identifying the risk of death due to progressive disease. Models of CanL that include aspects of the immune response will more accurately approximate natural disease evolution, which may offer an important tool for researchers targeting this deadly disease.

## 3 Materials and methods

### 3.1 cohort selection and diagnostics

A cohort of client-owned dogs naturally exposed to *L. infantum* in the Midwestern United States was selected based on a positive *Leishmania* diagnostic test or diagnostic positive dam or full sibling. Diagnostic tests included Real-Time quantitative Polymerase Chain Reaction (RT-qPCR) of *Leishmania* DNA from peripheral blood or Canine Visceral Leishmaniasis ® Dual Path Platform (DPP) serological test [18, 19]. Inclusion criteria included negative 4Dx Plus SNAP test (IDEXX Reference Labs) result indicating no recent exposure to tick-borne bacteria *Borrelia burgdorferi*, *Ehrlichia* spp., *Anaplasma* spp., or heartworm, and 2 or fewer outward clinical signs of CanL. Fifty dogs were enrolled in the study and assessed at three-month intervals over the course of 18 months. The cohort was block-randomized into two age and sex-matched groups where one group received a tick-preventative drug and the second group received a placebo. All dog caretakers gave signed informed consent, following a protocol approved by the University of Iowa Institutional Animal Care and Use Committee (IACUC).

### 3.2 Measurements of state components

At each study collection visit, pathogen load was quantified as parasite equivalents per mL of peripheral blood using RT-qPCR against a standard curve of blood derived from an unexposed dog spiked with known quantities of *L. infantum* promastigotes. To keep things simple, we focus here on a single infectious agent (*L. infantum*) and incorporate tick-borne bacterial co-infection as an indicator variable. Antibody levels were measured in dog serum via indirect enzyme-linked immunosorbent assay (ELISA). ELISA plates were coated with soluble *Leishmania* antigen (SLA) prepared from freeze-thawed stationary phase *L. infantum* promastigotes. The optical density (OD) ratio was calculated as the OD of test wells divided by a cutoff of the average OD plus 3 standard deviations of wells containing serum from unexposed negative control dogs. An OD ratio greater than 1 indicates a positive serological result. Disease status was determined using the LeishVet staging guidelines using complete blood count, serum chemistry panel, and physical examination information assessed by board-certified veterinarians [14].

Inflammatory and regulatory immune response variables were quantified from peripheral blood mononuclear cells (PBMCs) isolated by Ficoll-Paque PLUS density gradient. PBMCs were stimulated with 10 ug/mL total *Leishmania* antigen for seven days and then processed for

surface and intracellular staining with fluorophore-conjugated antibodies. Cell events were read on an LSR II flow cytometer and analyzed using FlowJo™ software [20]. Immune response readouts are presented as the percentage of cells positive for IFN-$\gamma$, IL-10, or PD-1 expression among CD4+ or CD8+ lymphocytes expressing high levels of CD49d, a marker of antigen-experienced T cells [21, 22]. Proliferation was measured using carboxyfluorescein succinimidyl ester (CFSE) dilution assay and presented as percent CFSE low cells among CD4+ or CD8+ CD49dhi lymphocytes.

### 3.3 Model specification: Longitudinal submodel

In Fig 1, we show an abbreviated illustration of the lag-1 temporal dependence structure of the Bayesian model, relating a given time index $t$ to the following time across the state components. We consider $N = 50$ dogs during the $T = 7$ time points. Mechanically, the state components are organized into a subject-specific row-vector $M_{i,t}$ of length 24, where $i = 1, 2, \ldots, N$ and $t = 1, 2, \ldots, T$. For the corresponding indexes $i$ and $t$, the row-vector $M_{i,t}$ includes terms representing the interplay of disease state with the rest of the model components: pathogen

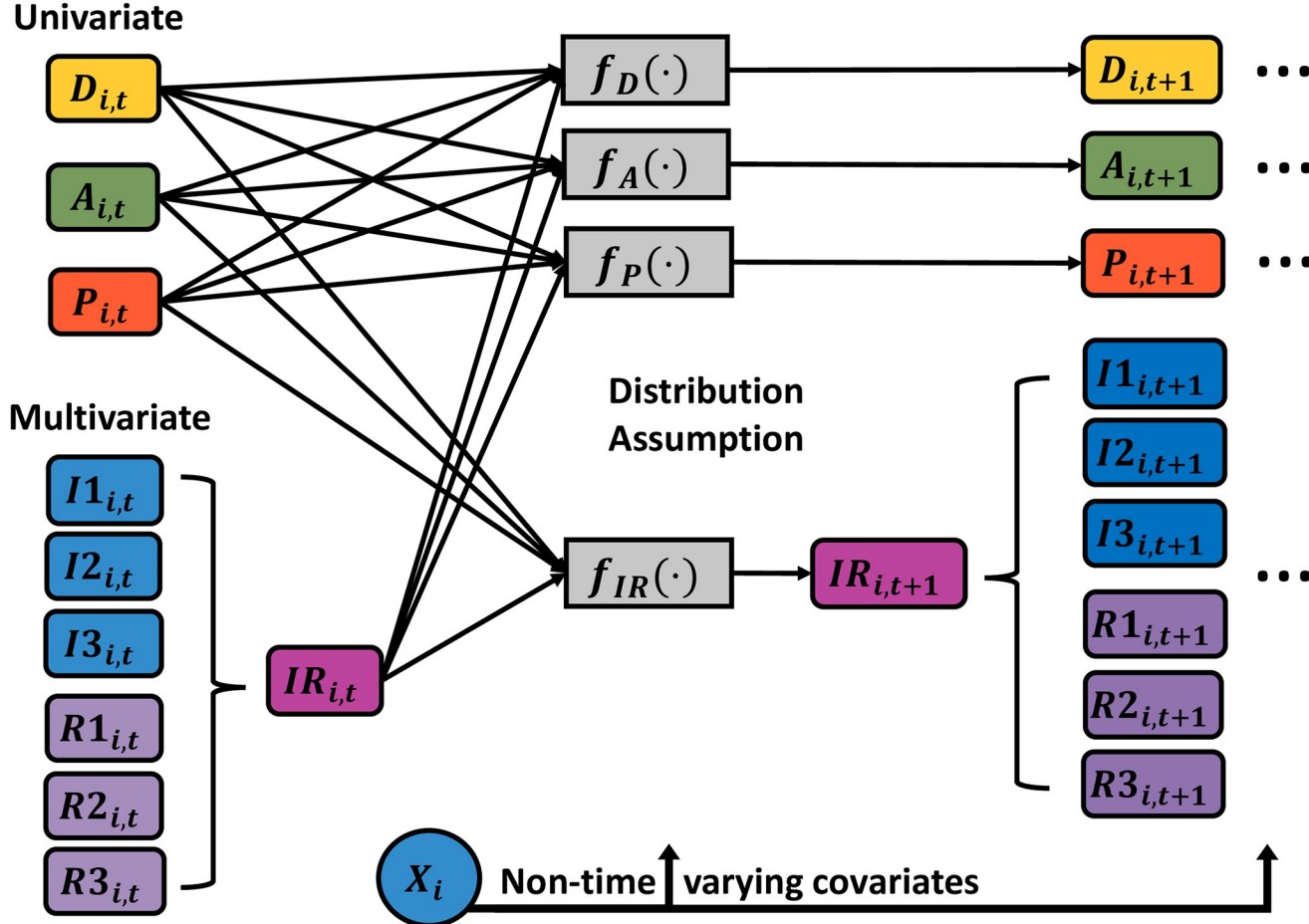

**Fig 1. Dynamic process.** Graphical display of how the disease state ($D$), antibody level ($A$), pathogen load ($P$), and inflammatory ($I$) and regulatory ($R$) responses are related to predicting the future value of each state for each dog $i$ at time $t + 1$. Non-time-varying predictors are organized into a subject-specific row-vector $X_i$. Expected future values of model components are determined based on corresponding functional forms $f(\cdot)$. For the case of inflammatory and regulatory responses, the term $IR$ corresponds to a vector of length 6 including the indicated immunologic responses.

load, antibody level, three inflammatory responses, and three regulatory responses. This specification allows us to assess the effects of the eight key drivers based on the dog's disease state. For clarity, we organize the non-time-varying predictors into separate subject-specific row-vectors $X_i$ of length 6, which include an intercept, indicators for age groups, 4Dx SNAP test result for tick-borne co-infection status, DPP result, and an indicator for tick prevention treatment. These two subject-specific row vectors drive the expected values of model components for each time point conditional on the previous time. This dependency occurs as a result of updated functions denoted by $f(\cdot)$ as presented in Fig 1. In this graphical display, we separate the univariate components of the model such as disease state, antibody level, and pathogen load, from those given a multivariate structure. Since there are known relationships between CD4 and CD8 T cells, all of the measurements coming from them could be potentially correlated. Thus, the inflammatory and regulatory responses are being modeled using a multivariate structure. Other specifications could consider a global multivariate vector autoregressive structure, but the chosen approach allows us to adapt the form of the model to known processes and relationships even in this relatively small sample setting.

To assess the CanL disease progression of subjects over time, we took into account LeishVet score, pathogen load, and level of anti-*Leishmania* antibodies as well as inflammatory and regulatory immune response elements, incorporated jointly in a longitudinal Bayesian submodel. To determine the clinical status of each dog based on professional veterinarian assessment and laboratory tests, the LeishVet scoring system [14] was used to classify the dogs into disease scores ranging from 0 to 4, with 0 indicating no signs of disease and 4 indicating severe disease. We then define less granular categories [3], based on LeishVet scores, consisting of four categories: healthy, mild, severe, and removed, as previously described by Pabon-Rodriguez et al. [3]. In this work, we go further and rewrite the disease state as a three-categorization outcome, where the removed state is now transformed into a whole new time-to-death submodel which is described in a subsequent section. The disease state variable is denoted by $D_{i,t}$ for the $i$th dog at time $t$, indicating the category into which the dog's state is classified. This categorization was then encoded through separate indicator variables for each score over time for each dog. Once an individual is infected with the parasite *L. infantum*, replication may occur. Concurrently, the host immune mounts a response against the parasite. To measure and quantify the pathogen status component of the overall dynamic process, we define $P_{i,t}$ to denote the pathogen load for the $i$th dog at time $t$. Further, let $A_{i,t}$ denote the anti-*Leishmania* antibody level for the same indexes. We denote the inflammatory state of an immune response by $I_{i,t}$ for $i$-th dog at time $t$. Since we consider several inflammatory signaling or responses, we then divided $I_{i,t}$ to define $I1_{i,t}$ to represent the proportion of IFN-$\gamma$ positive cells among CD4+ T cells. In a similar way, $I2_{i,t}$ and $I3_{i,t}$ are defined to capture the proportion of proliferating CD4+ and CD8+ T cells, respectively. The regulatory response $R_{i,t}$ components focus on CD4+ cells: IL-10 expression, and cell-surface PD-1 expression. As before, an example implementation of the regulatory state of an immune response might be separated into three components. First, $R1_{i,t}$ denotes the proportion of IL-10+ expressing CD4+ T cells, while $R2_{i,t}$ and $R3_{i,t}$ represent proportion of PD-1+ expressing CD4+ and CD8+ T cells, respectively.

Next, we present the Longitudinal Autoregressive Moving Average (ARMA) Bayesian model used for this work. Here, the AR component is used to model a linear combination of past time series values (lag-p values) and the MA piece for the linear combination of the past error (residuals) terms, in order to further improve predictions. Since each time point is equally spaced with 3 elapsed months, we decided to use an $ARMA(p, q)$ component of orders $p = 1$ and $q = 1$ for the continuous responses, which means that only the information and errors from the immediate previous time point are being used in the model. For the case of disease state, only an AR component of order $p = 1$ is considered. In addition, Fig 2 shows a

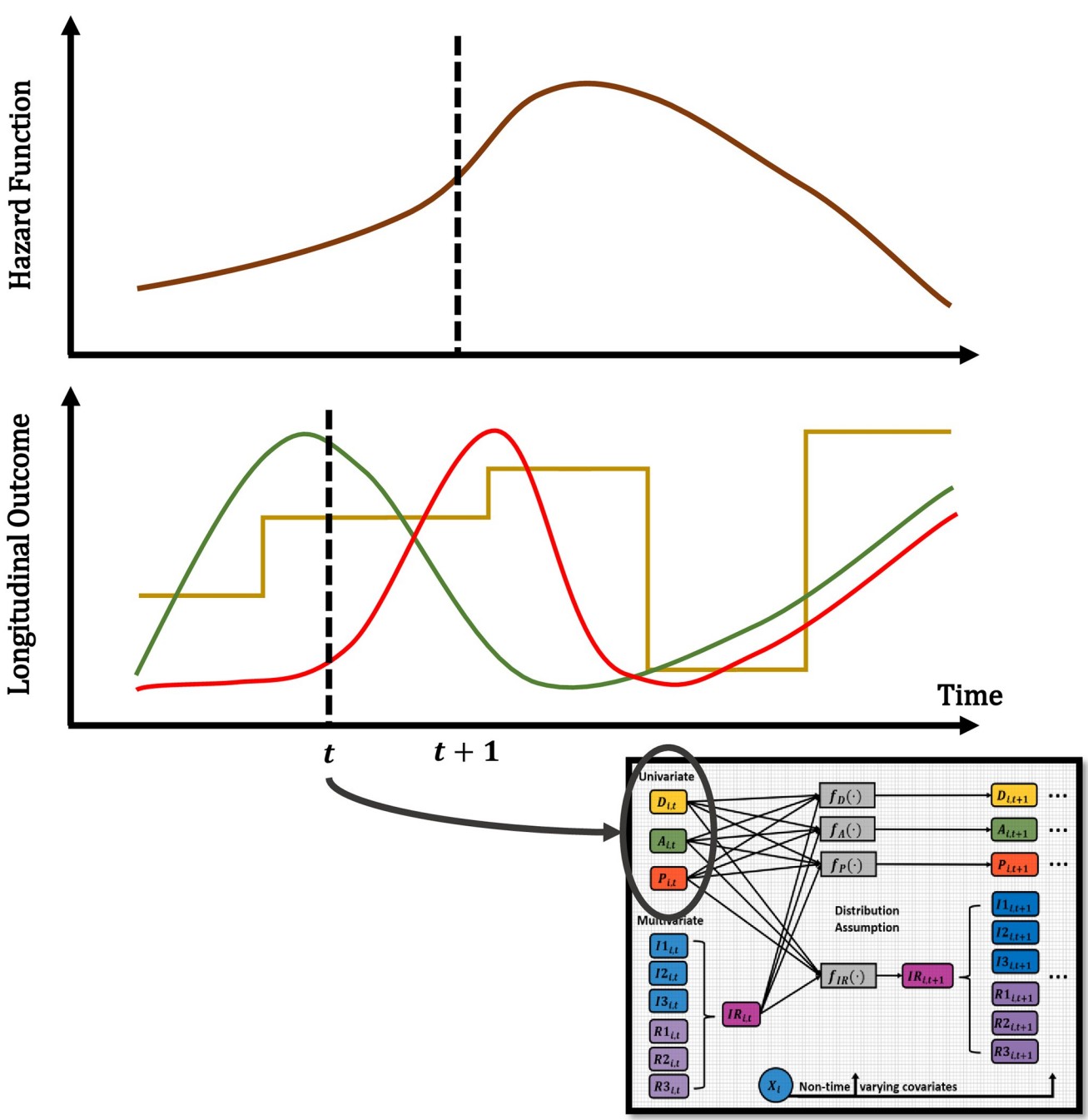

**Fig 2. Joint dynamic process.** Graphical display of how the longitudinal outcomes are structurally associated with the risk of death. A lagged structure is used to look at the association between the longitudinal outcome at the current time point ($t$) or the last observed time ($t^*$) into the risk of death (hazard) at the following time point ($t + 1$). The diagram shows an example of a possible univariate outcome and the hazard. The diagram serves as a visual representation, and the trajectories are not the actual data values.

graphical display of how the longitudinal outcomes are structurally associated with the risk of death. A lagged structure is used to look at the association between the longitudinal outcome at the current time point or the last observed time with the risk of death at the following time point. The risk of death due to progressive *Leishmania* infection is modeled through a survival

submodel via a proportional hazard specification with a Weibull baseline hazard function, with the goal of examining the association between longitudinal and survival processes.

**3.3.1 Pathogen load.**   When measures of pathogen load are obtained and calculated from qPCR results, there is always a possibility of readability or detection issues, particularly at lower levels of these detection signals. One of the most common problems when assessing results collected during a study is the presence of values below the limit of detection. For example, in this study, pathogen load was measured as the number of parasites per mL of blood (parasites/mL blood), and the detection of a small number of parasites was identified to be an issue, which led laboratory staff and researchers to use a Limit of Detection (LOD), which in this case was set to be 10 parasites per mL of blood ($LOD = 10$) and use a value of zero for pathogen concentration when not detected. It is important to determine the extent to which laboratory techniques can detect small quantities. Therefore, we say that a value is considered to be undetectable when it has crossed this particular limit of detection, which varies per instrumentation, and assay/test performed. The values below or above a limit of detection are considered to be censored values and are not missing at random in the usual sense because their absence reflects their value.

Based on Hornung and Reed [23], several techniques can be used in practice to model values below a limit of detection. For example, Hald [24] proposed a method of using maximum likelihood procedures based on a censored normal distribution, where the censoring point (limit of detection) is known and defined by the investigators. Another method is proposed by Nehls and Akland [25], which basically consists of setting all values below the limit of detection to be equal to $LOD/2$. One last but very simple approach is to set any value below the $LOD$ to be equal to zero. In this work, we model pathogen load for subject $i$ at time $t$ using a censoring mechanism and also transform the data in order to have a normally shaped distribution and improve the mixing of the model parameters. In this case, we scale pathogen load to bound the values to the interval $(0, 1)$ and modeled the logit of scaled pathogen load via a censored normal distribution as defined in Eq 1, where $\sigma_p$ is the standard deviation of the distribution. The logit function transforms the values into proportions, allowing a numerically stable computation. This function maps proportions to real numbers on the entire real number line.

$$
\begin{aligned}
logit(P_{i,t}) \quad &\sim \mathcal{N}_{cens}(\boldsymbol{M}_{i,t-1}\boldsymbol{\beta}_p + \boldsymbol{X}_i\boldsymbol{\alpha}_p + \theta_p w^p_{i,t-1}, \ \sigma_p) \\
w^p_{i,t} \quad &= logit(P_{i,t}) - (\boldsymbol{M}_{i,t-1}\boldsymbol{\beta}_p + \boldsymbol{X}_i\boldsymbol{\alpha}_p + \theta_p w^p_{i,t-1})
\end{aligned}
\tag{1}
$$

Here we have that $P_{i,t}$ is the scaled pathogen load, and $\boldsymbol{M}_{i,t-1}\boldsymbol{\beta}_p$ corresponds to the inner products of longitudinal outcomes and corresponding model coefficients (parameters), which are subject and time specific. The second term of the mean component is $\boldsymbol{X}_i\boldsymbol{\alpha}_p$, which is also an inner product including subject-specific covariates that are non-time dependent and their model coefficients. In this model component, the expression for the mean of the logit of scaled pathogen load, which is $\boldsymbol{M}_{i,t-1}\boldsymbol{\beta}_p + \boldsymbol{X}_i\boldsymbol{\alpha}_p + \theta_p w^p_{i,t-1}$, represents part of an ARMA specification, where the length of the parameters $\boldsymbol{\beta}_p$, $\boldsymbol{\alpha}_p$, and $\theta_p$ are 24, 6, and 1, respectively. Since the value $P_{i,t-1}$ is included in $\boldsymbol{M}_{i,t-1}$, then $\boldsymbol{M}_{i,t-1}\boldsymbol{\beta}_p$ represents part of the AR component while the term $\theta_p w^p_{i,t-1}$ represents the MA component. To represent the white noise (error) terms, we used the notation $w^p$ with corresponding subject and time indexes and the notation $\theta_p$ to represent the parameter accompanying the previous error terms, which are parts of the MA component. Finally, $\boldsymbol{X}_i\boldsymbol{\alpha}_p$ represents the constant term.

To characterize the censoring process, we treated all logit-scaled values below a corresponding transformed $LOD^* = logit(LOD) = logit(10)$ as left-censored and set them to be NA

("missing") and kept the values when they were above *LOD**, as defined in Eq 2.

$$logit(P_{i,t}) = \begin{cases} NA, & \text{if } logit(P_{i,t}^*) \leq LOD^* \\ logit(P_{i,t}^*), & \text{if } logit(P_{i,t}^*) > LOD^* \end{cases} \tag{2}$$

This censoring mechanism shown here is not the same censoring mechanism as commonly defined in the survival data analysis. The definition of left-censoring here is defined by the limit of detection issue present in the pathogen load, as explained at the beginning of this section. The values below a limit of detection are considered to be left-censored and set to missing.

**3.3.2 Antibody levels.** The anti-*Leishmania* antibody levels for the $i$th dog at time $t$, denoted by $A_{i,t}$, were measured by ELISA SLA OD ratio and then transformed using scaling and logit transformation as we did with pathogen load. In Eq 3, we model antibody levels with a logit-normal distribution with standard deviation $\sigma_a$.

$$\begin{aligned} logit(A_{i,t}) & \sim \mathcal{N}(\mathbf{M}_{i,t-1}\boldsymbol{\beta}_a + \mathbf{X}_i\boldsymbol{\alpha}_a + \theta_a w_{i,t-1}^a, \ \sigma_a) \\ w_{i,t}^a & = logit(A_{i,t}) - (\mathbf{M}_{i,t-1}\boldsymbol{\beta}_a + \mathbf{X}_i\boldsymbol{\alpha}_a + \theta_a w_{i,t-1}^a) \end{aligned} \tag{3}$$

As before, the white noise (error) terms are denoted by $w^a$ with corresponding subject and time indexes and the notation $\theta_a$ to represent the parameter accompanying the previous error terms.

**3.3.3 Disease status.** To assess and model disease progression, we used the LeishVet score proposed by Solano-Gallego et al. and kept the same variable definition to qualitatively describe the disease state of the subjects [7]. For $i$th dog at time $t$, we define disease state $D_{i,t}$ as in Eq 4.

$$D_{i,t} = \begin{cases} 1 \,(\text{Healthy}), & \text{if } LeishVet = 0 \text{ or } 1 \\ 2 \,(\text{Mild}), & \text{if } LeishVet = 2 \\ 3 \,(\text{Severe}), & \text{if } LeishVet = 3 \text{ or } 4 \end{cases} \tag{4}$$

For modeling purposes, we use a multinomial-logit link specification for a disease state, as shown in Eq 5. Here, $\pi_{i,t}^{(k)}$ defines the probability that the $i$th dog at time $t$ is classified with disease state $k = 2, 3$. For instance, the probability of a subject being in the "healthy" category is then denoted within the multinomial specification by $\pi_{i,t}^{(1)}$ at given indexes and computed based on the other probabilities.

$$\begin{aligned} D_{i,t} & \sim \text{Multinomial}(1; \ \pi_{i,t}^{(1)}, \ \pi_{i,t}^{(2)}, \ \pi_{i,t}^{(3)}) \\ \pi_{i,t}^{(k)} & = \frac{exp[\mathbf{M}_{i,t-1}\boldsymbol{\beta}_d^{(k)} + \mathbf{X}_i\boldsymbol{\alpha}_d]}{1 + \sum_{g=2,3} exp[\mathbf{M}_{i,t-1}\boldsymbol{\beta}_d^{(g)} + \mathbf{X}_i\boldsymbol{\alpha}_d]}; k = 2, 3 \\ \pi_{i,t}^{(1)} & = 1 - \pi_{i,t}^{(2)} - \pi_{i,t}^{(3)} \end{aligned} \tag{5}$$

The multinomial-logit link specification for disease state presented in this current work differs from the one proposed by Pabon-Rodriguez et al. [3]. In the current work, we removed the last category and added a separate survival submodel to account for those subjects that were censored or died due to severe CanL. For this, we used post-follow-up time points after the study ended, corresponding to time points 8 and 9, where additional deaths were reported due to the disease or unrelated reasons. In addition, we use the "healthy" category as the

reference level rather than the last category since we want to compare advanced disease states to healthy subjects.

**3.3.4 Inflammatory and regulatory responses.** To further improve understanding of CanL progression based on immune responses, longitudinal inflammatory and regulatory immune outcomes were taken into consideration. Since there are known relationships between CD4 and CD8 T cells, all of the measurements coming from them could be potentially correlated. To account for this, we constructed a multivariate component in the model for these responses as presented in Eq 6, where for each dog and time-point we have that $IR_{i,t} = [I1_{i,t}\ I2_{i,t}\ I3_{i,t}\ R1_{i,t}\ R2_{i,t}\ R3_{i,t}]'$ of dimension 6×1. Since the raw values for these components were already expressed in decimal forms in the interval (0, 1), we only used a logit transformation and modeled them as specified.

$$
\begin{aligned}
logit(\boldsymbol{IR}_{i,t}) &\sim MVN_6(\boldsymbol{\Theta}_{i,t}^{(IR)}, \boldsymbol{\Sigma}_{i,t}^{(IR)}) \\
\boldsymbol{\Theta}_{i,t}^{(IR)} &= (\mathbb{I}_6 \otimes \boldsymbol{M}_{i,t-1})\boldsymbol{\beta}_{ir} + (\mathbb{I}_6 \otimes \boldsymbol{X}_i)\boldsymbol{\alpha}_{ir} + (\boldsymbol{T}_{ir})\boldsymbol{w}_{i,t-1}^{ir} \\
\boldsymbol{\Sigma}_{i,t}^{(IR)} &= \boldsymbol{\Sigma}^{(IR)} \\
\boldsymbol{w}_{i,t}^{ir} &= logit(\boldsymbol{IR}_{i,t}) - \boldsymbol{\Theta}_{i,t}^{(IR)}
\end{aligned}
\tag{6}
$$

The parameters associated with the MA component in this multivariate structure are $\boldsymbol{w}^{ir}$, a vector of length 6 representing the white noise (error) terms, and $\boldsymbol{T}_{ir}$ a 6×6 matrix of parameters for previous error terms. For each pair of subject and time indexes, we have that $\boldsymbol{\Theta}_{i,t}^{(IR)}$ is the mean vector of length 6, and $\boldsymbol{\Sigma}_{i,t}^{(IR)} = \boldsymbol{\Sigma}^{(IR)}$ the variance-covariance matrix of dimension $6 \times 6$.

## 3.4 Model specification: Survival submodel

To measure the impact of longitudinal outcomes on the risk of death due to progressive Leishmaniosis, a survival submodel is defined to examine the association between longitudinal outcomes, predictors, and the risk of death due to progressive disease. Here, the survival time for individual $i$ is modeled from a proportional hazard specification with a Weibull baseline hazard function $h_0$ as shown in Eq 7.

$$
h_i(t|\cdot) = h_0(t)exp\left(\boldsymbol{X}_i\gamma + \sum_{j=1}^{8}\omega_j m_{i,j}(t^*)\right),
\tag{7}
$$

This submodel requires the definition of an association structure between the longitudinal and survival data, an important component in joint modeling [26]. In this case, the association structures, denoted by $m_{i,j}(t^*)$, use a lagged effect property. They are defined as the expected value of the longitudinal outcome $j$ at the last observed time $t^*$ of subject $i$. The association parameters $\omega$'s quantify the associations between each longitudinal outcome and the risk of death due to severe disease, and $\gamma$'s examine the effect of the non-time-varying covariates. For instance, since pathogen load is the first component we explained in Subsubsection 3.3.1, let us assume $j = 1$, then $m_{i,1}(t^*) = E[logit(P_{i,t^*})]$, and $w_1$ measures the association of pathogen load on the risk of death. While the longitudinal submodel uses information on time points from TP1 to TP7, the survival submodel on the other hand, uses information from TP1 to TP7 as well as two post-follow-up time points after the study ended (TP8 and TP9) where more deaths due to severe cases of diseases and unrelated to it were identified.

### 3.5 Computation and model diagnostics

To fit the presented Bayesian model, we used the `NIMBLE` [27] package in R [28, 29] along with the `parallel` [28] package to obtain posterior samples of model parameters and latent quantities using Markov Chain Monte Carlo (MCMC) techniques. A general description of MCMC techniques is provided by Liu [30] and Carlin [31]. In this case, due to the selection of priors and definition of the model components, default samplers from `NIMBLE` seemed appropriate and hence were used via the configuration tool provided in the package.

In Bayesian analysis, the prior distribution represents the knowledge or belief about the parameters before the data is observed. The choice of the prior distribution and its parameters can have a significant impact on the posterior distribution, which is the updated knowledge or belief about the parameters after the data is observed. Independent normal prior distributions with zero mean, $\mathcal{N}(0, \sigma_\beta^2)$, were used for the conditional and unconditional effects ($\boldsymbol{\beta}$'s) in order to shrink effects towards zero, where gamma prior distributions $\Gamma(1, 1)$ was used for the variance of the effects ($\sigma_\beta^2$), which is a hyperparameter. All of the parameters associated with the non-time-varying predictors ($\boldsymbol{\alpha}$'s), the moving-average parameters associated with pathogen load, and antibody levels ($\theta_p, \theta_a$) use independent standard normal prior distributions $\mathcal{N}(0, 1)$. This choice of prior indicates that there is no strong prior information or belief about the magnitude or direction of the effect of the non-time-varying predictors or moving-average parameters on the outcomes, and allows the data to drive the posterior inference.

Elements of the matrix $\boldsymbol{T}_{ir}$ uses independent normal prior distributions $\mathcal{N}(0, 1)$. A gamma prior distribution $\Gamma(1, 1)$ was also used for the variance terms of the model components ($\sigma_a^2$, $\sigma_p^2$). The prior for the covariance matrix $\boldsymbol{\Sigma}^{(IR)}$ was chosen to be an Inverse Wishart, $\mathcal{W}^{-1}((N - p - 1)\mathbf{S}, N)$, where $N$ is the number of subjects, while $p$ and $\mathbf{S}$ are the dimensions and the sample covariance of the Inflammatory-Regulatory responses, respectively. The white noise ($w$'s) or residual terms of the continuous responses are derived parameters within the model. In the case of the survival submodel, the association parameters ($\omega$'s) assumed standard normal prior distributions $\mathcal{N}(0, 1)$, while the covariate effects ($\gamma$'s) use independent normal prior distributions with zero means and each with different variance term with a gamma prior distribution $\Gamma(1, 1)$.

The normal distribution is a commonly used prior distribution for model parameters because it is mathematically convenient and flexible. While the normal distribution is a popular choice for prior distributions, it is not always appropriate or necessary. Some reasons that led us to use this distribution are:

(i). Conjugacy: The normal distribution is a conjugate prior for the parameters of many commonly used likelihood functions. This means that the posterior distribution will also be a normal distribution, making computation of the posterior distribution relatively straightforward.

(ii). Flexibility: The normal distribution is a flexible distribution that can be parameterized to allow for varying degrees of prior information or uncertainty. Specifically, the mean and variance parameters can be used to represent the prior mean and precision (or equivalently, the prior standard deviation) of the parameter values.

(iii). Familiarity: The normal distribution is a well-known and widely used distribution, making it familiar to many researchers and practitioners. This can make it easier to communicate results and compare findings across studies.

Among the hyperparameters within the presented Bayesian model, the variance parameter of the prior distribution for the conditional and unconditional effects as well as for the

association parameters controls the amount of information or uncertainty that is incorporated into the prior distribution. A smaller variance indicates a more informative prior, where the prior distribution is concentrated around a particular value, indicating a stronger prior belief in the parameter value. A larger variance indicates a less informative prior, where the prior distribution is more spread out, indicating weaker prior belief in the parameter value. By varying the variance parameter of the prior distribution, you can explore the impact of different levels of prior information or uncertainty on the posterior distribution. A sensitivity analysis can help us determine how sensitive the posterior distribution is to changes in the variance parameter and can inform decisions about the appropriateness of the prior distribution and the level of prior information or uncertainty to use in the analysis.

In the current work, we performed a sensitivity analysis (see supplementary file S2 File), targeting the prior information on the variance of the model parameters, particularly on the driver effects and association parameters between the longitudinal and survival submodels. For variance parameters (or standard deviations), we used gamma-distributed priors as previously discussed. Thus, we varied the shape and scale parameters of the gamma distribution to create different levels of prior information or uncertainty. Specifically, we considered the following scenarios with different levels of prior variance:

(i). Gamma distribution: The first scenario uses the $\Gamma(1, 1)$ distribution mentioned above. This results in a prior distribution that is weakly informative and places a lot of mass near zero but allows for a wide range of possible values. This prior can be useful when you have very little information about the variance and want to avoid overfitting by not allowing the variance to become too large.

(ii). Uniform distribution: This scenario considers a uniform distribution to model prior uncertainty about the coefficient variance across the entire range of possible values. We used a $U(0, 100)$ distribution, a relatively uninformative prior, which assumes equal probability for any value of the variance between 0 and 100. We assumed no prior knowledge or strong beliefs about the possible range of values for the variance.

(iii). Inverse Gamma distribution: For the third scenario, we specified a $\Gamma^{-1}(2, 0.5)$, which can be considered an informative prior for the variance parameters. This would put more prior weight on larger values of the variance compared to the previous scenarios but still allow for a wide range of possible values. The shape parameter of 2 indicates some prior belief that the variance is not very small, while the scale parameter of 0.5 indicates some prior belief that the variance is not very large.

For the case of unobserved or missing longitudinal outcomes (latent quantities), we used Bayesian imputation within the analysis to obtain the posterior distribution of these quantities of interest. In Bayesian approaches for longitudinal data, we assume that the missing data are missing at random (MAR), which means that the probability of missingness depends only on observed data, and not on the missing values themselves after conditioning on the observed data. On the other hand, in the survival submodel, missing data can occur when an individual is lost to follow-up or drops out of the study before the end of the follow-up period. The probability of being lost to follow-up or dropping out may depend on observable characteristics, such as age or sex, or unobservable characteristics, such as health status or treatment response. When missing data occur due to unobservable characteristics, the data are said to be missing not at random (MNAR), since the missingness mechanism is related to the unobserved outcomes. While MNAR and MAR are mutually exclusive assumptions in general and cannot occur at the same time, in this particular case of the joint modeling of longitudinal and survival data, it is possible to have both. This is because the missingness mechanism can be different

for the longitudinal and survival data components. For example, the missingness in the longitudinal data may depend on the observed longitudinal data and some additional unobserved covariates, while the missingness in the survival data may depend on the observed survival data and some other unobserved covariates. In this case, the longitudinal data would be MAR with respect to the observed data and the unobserved covariates, while the survival data would be MNAR with respect to the unobserved covariates.

Posterior results are based on a set of three (3) MCMC chains of 20, 000 iterations on each chain, after the burn-in period. We ran the Bayesian model on a laptop computer with an AMD Ryzen 9 processor and 24 GB of memory in order to evaluate the computing performance. We achieved a running time of around 13 hours for the number of iterations per chain ran in parallel, which is shorter than the running time obtained in Pabon-Rodriguez et al. [3]. While the running time was reduced in the current work for a more complex model specification, we still believe there are numerous opportunities for optimization.

To assess the convergence of the parameters, a Gelman-Rubin diagnostic was computed using the `coda` package [32] in R. As a rule to assess convergence, any potential scale reduction factor below 1.2 indicates that the corresponding parameter has reached a level of convergence. In this case, all parameters achieved an estimated factor below the chosen threshold of 1.2. We also computed the Monte Carlo Standard Error (MCSE), which is a measure of the precision of the posterior distribution obtained from an MCMC algorithm. The purpose of MCSE is to provide a measure of the accuracy of the estimate of the posterior distribution, which is the distribution of the parameters of interest after taking into account the data and prior information [33]. For the parameters in the model, we obtained MCSE values between 0.00033 and 0.06583, with a mean of 0.00568 and a median of 0.00328. The full MCMC summary and diagnostics can be found in the supplementary file S1 File.

The code implementing the sampler using the `NIMBLE` system, and the posterior summary, diagnostics, sensitivity analysis, and code for the forecasting are provided in the supplemental material in the GitHub repository.

## 3.6 Posterior predictive distribution and p-values

The posterior sampling process (simulation) was used to obtain posterior predictive trajectories. The idea is to use the posterior distribution of the model parameters and the model specification provided throughout previous subsections. The posterior predictive trajectories are obtained by following the steps:

(i). Sample a value from the posterior distributions of each model parameter,

(ii). Obtain the estimated model components (predicted outcomes) from the Bayesian model specification from Eqs 1 to 6,

(iii). Repeat steps (i) and (ii) for $S = 1000$ times ($s = 1, 2, 3\ldots, S$), where $S$ represents the number of simulations performed, and

(iv). Plot the resulting posterior predictive trajectories.

Posterior predictive p-values are a Bayesian statistical method used to assess the fit of a model to observed data [31]. They measure the probability of observing data as extreme or more extreme than the observed data, given the model and the prior distribution. In the Bayesian framework, a posterior predictive p-value is an important tool for model checking and selection [33]. It allows us to evaluate whether the model adequately explains the observed data and whether the model is consistent with prior knowledge and assumptions. This approach can help identify potential flaws in the model and guide the choice of alternative models. To

compute the posterior predictive p-value, a test statistic is required to summarize the observed data [33]. This test statistic is then compared to a distribution of the same test statistic generated from samples from the posterior predictive distribution. In the current work, we used the mean, standard deviation, and mean squared error (MSE) as test statistics.

To obtain the posterior predictive p-values, we followed these steps:

(i). Fit a Bayesian model to the data and obtain samples from the posterior distribution using MCMC methods

(ii). Generate new data from the posterior predictive distribution using the sampled parameters from the posterior distribution as described at the beginning of this subsection

(iii). Compute the test statistic on the observed data and on each simulated dataset from the posterior predictive distribution

(iv). Calculate the p-value as the proportion of simulated test statistics that are as extreme or more extreme than the observed test statistic

## 4 Results

In this analysis, we utilized the LeishVet clinical scoring scale and the more adaptive qualitative information, as shown in Eq 4. A similarly transformed criterion was first presented in Pabon-Rodriguez et al. [3] with the purpose of distinguishing between healthy dogs and those exhibiting mild or severe cases of the disease. This definition led us to define a multinomial distribution for disease status and make use of basic definitions to infer and interpret our results. For instance, it is known that the sum of the probabilities of the clinical score categories should be equal to 1. We used this property to infer that if the probability of the first disease score category decreases by a certain amount, the probability of disease progression is evident, because either one of the remaining categories' probabilities increases, indicating a more advanced case of the disease.

A table summarizing model parameters is provided as supplemental material as well as in the GitHub page provided in Section 3.5. This table presents a summary of the posterior results for all parameters in the model, including posterior means, median, standard deviation (SD), 95% credible intervals (Cr-I), and the posterior probabilities of each parameter ($x$) being either positive or negative, which is $P(x > 0)$ or $P(x < 0)$. This is the main criterion used for assessing the strength of evidence for corresponding parameters. The threshold value for the posterior probability is set to be 0.65. To determine whether the effect is moderate or strong, we use the following rule: We consider any parameter with a posterior probability of being positive (or negative) in the interval [0.65, 0.85) to be a moderate effect, while any parameter with a posterior probability of being positive (or negative) in the interval [0.85, 1] will be considered as showing a strong effect. Using this approach, we determine which parameters drive disease progression among the cohort and its dependence on future occasions. Moderate and strong associations between model parameters are summarized graphically in Fig 3.

The trajectories of observed pathogen loads and antibody levels over each of the seven time points, each separated by three months, are shown in Fig 4. In addition, we show the trajectories of individual immune variables, each measured as proportions in decimal form.

Concerning pathogen load, we found its dependence on previous time point values varied with disease stage as measured by the LeishVet score. For example, the conditional interaction effect on lag-1 pathogen load (previous time period's pathogen load) for score 1 dogs had a posterior mean of 0.409 with 95% Cr-I [0.211, 0.597] and posterior probability 0.999 that the true parameter is greater than zero. We also found strong evidence that antibody levels in

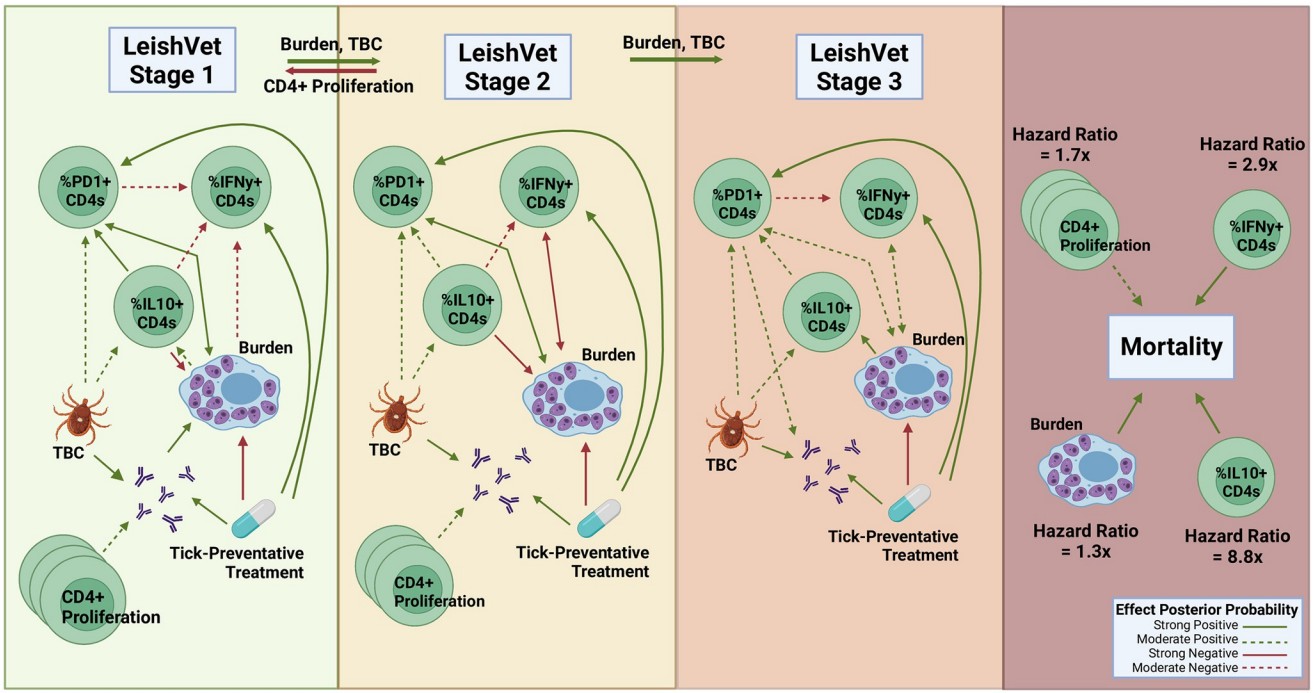

**Fig 3. Strength associations during CanL disease progression.** Graphical representation of associations between model parameters based on the estimated posterior probabilities of the effect being either positive (represented by green lines) or negative (represented by red lines). We consider any parameter with posterior probability in the interval [0.65, 0.85] to be a moderate effect (represented by a dashed line), while any parameter with posterior probability in the interval [0.85, 1] will be considered as showing a strong effect (represented by a solid line). TBC—Tick-borne pathogen co-infection. Figure created using BioRender.com.

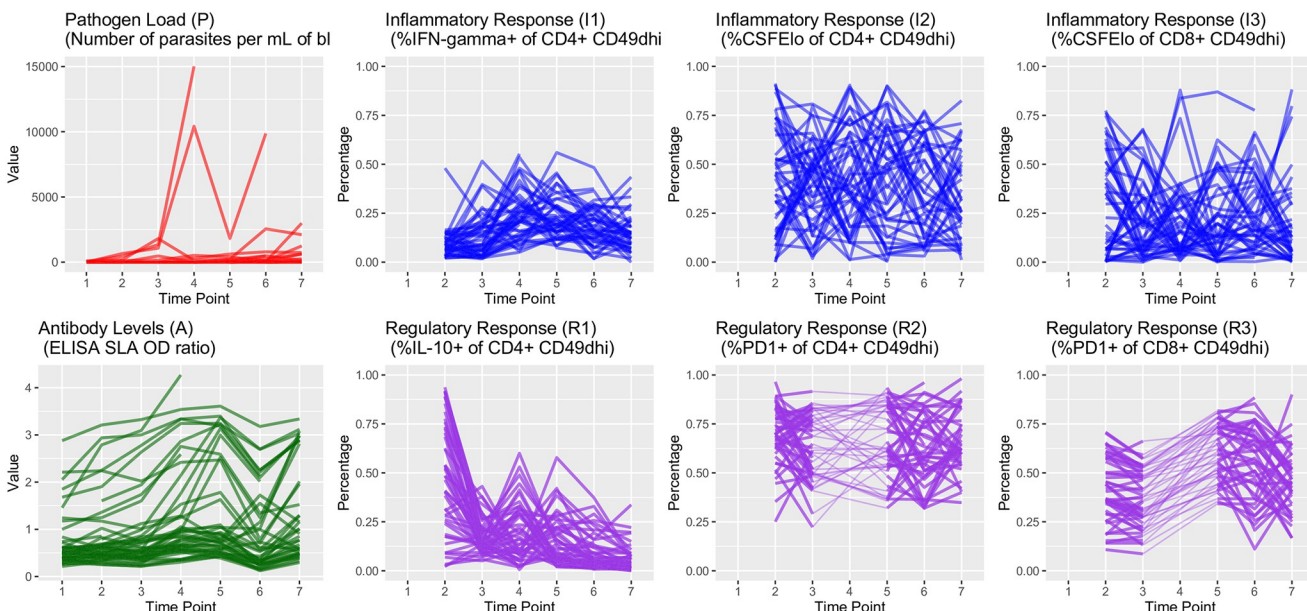

**Fig 4. Longitudinal plots of continuous outcomes.** The first plot in the upper row shows *Leishmania* pathogen load over time, expressed as the parasites per mL of blood, and the inflammatory responses (in blue) are expressed in terms of percentage of expression. In the second row, we have antibody levels expressed as the ratio to control cut-off (in color green), and the regulatory responses (in purple) expressed as a percentage of expression. Each line represents a single dog's trajectory over the study period. Each time point corresponds to 3 elapsed months. For the inflammatory and regulatory responses, the information on time point 1 is unobserved (latent). In addition, information in time point 4 for the responses labeled as R2 and R3 is also unobserved. Observed information from time points 3 to 5 is connected via dashed lines to retain subject trajectory information.

LeishVet score 1 dogs were associated with higher pathogen load, as its posterior mean was 3.374 with 95% Cr-I [2.442, 4.399] and posterior probability 1.00 of being positive. Petersen et al. [34] showed that when the inflammatory immune response is active, the parasite burden is diminished. We found that immune response activity in LeishVet score 2 dogs had the strongest probability of affecting future pathogen load. Percent IFN-$\gamma$+ CD4+ T cells produced a negative effect on the pathogen load in LeishVet score 2 dogs. The posterior probability of this effect being negative is 0.897, where the posterior mean of the effect is −0.483 [−1.240, 0.287]. Also in LeishVet score 2 dogs, percent IL-10+ CD4+ T cells had a strong negative effect on future pathogen load (posterior probability of negative effect of 0.970). Interestingly, considering non-time-varying predictors, we found evidence that the group receiving tick preventative treatment had lower overall parasite loads (posterior probability of being negative of 0.892), with posterior mean of −0.550 with 95% Cr-I [−1.432, 0.324].

We examined possible drivers of increased antibody levels and found the percent proliferating CD4+ T cells had moderately positive estimated effects on antibody levels in both LeishVet score 1 and score 2 dogs. The posterior probabilities for these two effects are 0.769 and 0.777 of being positive, respectively. Percent PD1+ CD4+ T cells also had a positive effect on antibody levels within LeishVet score 3 dogs (posterior probability of being positive of 0.739). Regarding non-time-varying predictors, we found strong evidence that exposure to tick-borne co-infections (indicated by positive 4Dx Plus SNAP test result) had a positive effect on anti-*Leishmania* antibody levels, with a posterior probability of being positive of 0.987. This effect has a posterior mean of 0.157 with 95% Cr-I [0.017, 0.306]. Furthermore, higher antibody levels were related to stronger DPP readings (posterior probability 0.999) and tick preventative treatment (posterior probability of being positive 0.930).

Using this model approach, we were able to identify parameters associated with the probabilities of dogs maintaining disease states and which factors drive increasing disease severity. As expected, pathogen load had a strong positive effect on the probability of LeishVet score 1 dogs progressing to score 2 (posterior probability 0.997), as well as the probability of dogs progressing from LeishVet score 2 to score 3 (posterior probability 0.926). We also found that percent proliferating CD4+ T cells showed a strong negative effect on disease progression, reducing the probability that LeishVet score 1 dogs progress to LeishVet score 2. The posterior probability for this effect was 0.958. This analysis also captured the interesting effect that tick vector-borne co-infection status had a strong positive effect on disease progression with posterior probabilities of 0.986 and 0.830 regarding LeishVet scores 2 and 3, respectively.

Several interesting signals were found when we analyzed immunological parameter effects within the model. In LeishVet score 1 dogs, previous percent PD-1+ of CD4+ T cells identified with a decreased effect on future percent IFN-$\gamma$+ CD4+ T cell values, with a posterior probability of being negative of 0.782. This decreased effect was also found in LeishVet score 3 dogs (posterior probability of being negative of 0.736). Pathogen load was consistently, positively, associated with increased percent PD-1+ of CD4+ T cells at LeishVet score 1, 2, and 3 (posterior probability 0.965, 0.967, and 0.778 respectively). Pathogen load reduced future percent IFN-$\gamma$+ CD4+ T cell values during the lower disease score states (posterior probabilities 0.713 and 0.689). At LeishVet score 3, however, pathogen load was associated with increased future percent IFN-$\gamma$+ CD4+ T cell values (posterior probability 0.736). Increasing parasite burden produced a positive effect on future percent IL-10+ CD4+ T cell values among LeishVet score 1 dogs (posterior probability 0.826) and LeishVet score 3 dogs (posterior probability 0.856). We found that percent IL-10+ CD4+ T cell values were also associated with decreased percent IFN-$\gamma$+ CD4+ T cell at lower disease scores (posterior probabilities 0.740 and 0.774), which may represent a feedback mechanism between the two cytokines. In addition, across the disease score range, we found that percent IL-10+ CD4+ T cells was associated with increasing

percent PD-1+ CD4+ T cells, where the posterior probabilities were 0.903, 0.724, and 0.757, for LeishVet score 1, 2, and 3, respectively.

Regarding non-time-varying predictors, we found tick preventative treatment had strong positive effects (posterior probability of 0.910) on future percent IFN-$\gamma$+ CD4+ T cells and percent PD-1+ CD4+ T cells (posterior probability of 0.991). Tick-borne pathogen co-infection status showed positive effects on percent IL-10+ and PD-1+ CD4+ T cells, with posterior probabilities of being positive of 0.806 and 0.786, respectively.

The longitudinal submodel has allowed us to identify evidence of possible drivers of CanL progression by disease classification, but we can also look at the association of the longitudinal outcomes with the risk of death due to progressive CanL within the cohort. This association is possible via the survival submodel, as illustrated in Fig 2 and explained by Eq 7, where a proportional hazard specification is used with a Weibull baseline hazard function, and a lagged-effect structure. We found a strong association that lag (previous value of) pathogen load increases the hazard of death due to progressive CanL. The posterior mean of this association is 0.270 with 95% Cr-I [−0.020, 0.600] with a posterior probability of a positive association of 0.965. This association value translates into a hazard ratio (HR) of 1.310. Concerning immune response parameters, we found that both lag percent IL-10+ CD4+ T cell values (associated with regulatory responses), as well as percent proliferation among CD4+ T cells and percent IFN-$\gamma$+ CD4+ T cells (associated with inflammatory responses), are among the responses with a strong positive association with the risk of death due to progressive CanL (posterior probabilities of 0.999, 0.770, and 0.933, respectively). These association values translate into HRs equal to 8.791, 1.674, and 2.930, respectively based on mean values, which indicates a surprisingly high risk of death associated with these indicated immune responses, especially percent IL-10+ CD4+ T cell values. In terms of non-time-varying predictors, we found that younger subjects showed a moderate positive association with the risk of death due to progressive CanL (posterior probability of 0.663).

## 4.1 Posterior predictive trajectories

We examined the extent to which the presented Bayesian model could forecast trajectories of CanL progression. We considered four dogs within the cohort showing outcome patterns of decreasing, stable, or growing clinical status. One disadvantage of longitudinal forecasts for dynamic processes generally is that predictions further forward in time increase model uncertainty. Accordingly, we assumed only data from the first two time points were available and used a six month forecast window, which is a clinically relevant time frame for this disease. All of the posterior predicted trajectories are presented in the supplementary figures from S1 to S9 Figs. In addition, we only provide the 95% credible bounds of the posterior predictive trajectories. Here, we noticed that the posterior predictive intervals are wider when looking at time point 4 (six month forecast window) compared to the narrower intervals obtained when predicting responses for time point 3 (three month forecast window).

As discussed in Subsubsection 3.3.1, pathogen load was defined based on a censoring approach, therefore the model should be able to detect a trajectory of censored values. S1 Fig shows the posterior predictive trajectories of the pathogen load of the four example subjects. Subjects 2, 3, and 4 had censored values, and the model generated a median predicted trajectory similar to the observed trajectory at time point 3, but a slight increase is shown in pathogen load from time point 3 to 4, which was accompanied by a more notable change in antibody production as shown in S2 Fig. This caused a change in the probabilities of disease state classification as shown in S3 Fig. At the second time point, subject 2 is in LeishVet score 1 with probability 1, but the probability of remaining score 1 decreased gradually, while the

probability of progressing to LeishVet score 2 or 3 increased. The increase in pathogen load and antibody levels in this subject caused deterioration in the probability of improving clinical status, as seen in the graph, when the probability of LeishVet score 1 decreases from 1 at time point 2 to 0.70 by time point 4, while the marginal probability of LeishVet score 2 for this subject is at approximately 0.20 by time point 4. In the case of subject 3, which is in LeishVet score 2 with probability 1 at time point 2, its marginal probability of LeishVet score 2 is reduced to around 0.45, while the marginal probability of clinical improvement to LeishVet score 1 disease is around 0.41 at time point 4. Subsequent graphs in the supplementary figures S4 to S9 Figs show posterior predictive trajectories of immune components, where S4–S6 Figs correspond to responses associated with the inflammatory immune response, while S7–S9 Figs are associated with the regulatory immune response. Among these, the model performed well in predicting the proportion of IFN-$\gamma$+ and IL-10+ CD4+ T cells, which are of great importance for immunopathogenesis during CanL.

The posterior predictive p-values for pathogen load and antibody levels readouts are 0.242 and 0.486, respectively. For the case of the inflammatory and regulatory responses of the immune system, we obtained the posterior p-values: 0.348, 0.868, 0.563, 0.801, 0.494, and 0.847. These posterior p-values correspond, respectively, to the readouts as presented in Fig 4. While there is no universal rule of thumb for posterior p-values, on many occasions, a common approach considers a posterior p-value close to 0.50 to indicate that the observed data is perfectly consistent with the model and prior assumptions, but other research works have also used the rule of a p-value between 0.15 and 0.85 to be acceptable. It is worth noting that the interpretation of a posterior p-value can also depend on the sample size, as larger samples may lead to smaller posterior p-values even if the data are relatively consistent with the model and prior assumptions. In this work, we handled a small sample setting, which could lead to larger posterior p-values. Based on the estimated posterior p-values, we can say that our observed data are relatively consistent with the model and prior assumptions.

## 5 Discussion

Clinical management of CanL is difficult due to its intricacy and a wide array of clinical signs, which have a high degree of overlap with other ailments and common infections. The LeishVet group proposed a method for classifying clinical CanL [14] based on a combination of clinical signs, clinicopathological laboratory abnormalities, and serological status. The staging system can aid clinicians in identifying proper therapy, anticipating prognosis, and executing follow-up actions. We hypothesize modeling disease status as dependent on multiple parameters including pathogen load, antibody level, and inflammatory and regulatory immune factors will improve model predictions and expand model applications. Through this novel statistical analysis, we were able to characterize interactions between specific pathogen and host immune response parameters and assess possible drivers of CanL disease progression.

In the Bayesian model presented here, we examined pathogen load as a longitudinal outcome possibly affected by immune parameters and found several interesting relationships, summarized in Fig 3. The model suggests induction of IFN-$\gamma$+ CD4+ T cells during the lower scores of the disease is associated with a decrease in pathogen load. This is consistent with the known function of IFN-$\gamma$+ to activate macrophage microbicidal responses. Interestingly, percent IL-10+ CD4+ T cells were also associated with decreased pathogen load in lower disease score stages, this could indicate that the initial CD4+ T cell response is comprised of cells co-producing IFN-$\gamma$+ and IL-10, where the relative efficacy of IFN-$\gamma$+ to IL-10 is sufficient to exert inhibition of parasite growth. CD4+ T cells with this Type 1 regulatory cell phenotype have been observed in experimental visceral leishmaniasis [35]. However, IL-10+ CD4+ T cells

may indirectly promote increased parasite burdens through a positive association with increasing PD-1+ CD4+ T cells observed at LeishVet scores 1, 2, and 3. Percent PD-1+ CD4+ T cells were associated with increased pathogen load throughout the course of CanL disease scores, which may indicate an important role for PD-1 in influencing parasite control. In agreement with our findings, it has been shown that blocking the interaction of PD-1 with its ligand, PD-L1, among canine peripheral mononuclear cells or splenocytes, enhances leishmanicidal activity [1, 36]. PD-1 immunotherapy has been proposed as a potential treatment strategy for treating dogs with CanL [37], the results of the model presented here support PD-1 as a therapeutic target for reducing parasite load in dogs with CanL.

PD-1 is upregulated on T cells following normal activation by cognate antigen-presenting cells in order to temper T cell responses. Therefore, PD-1 can serve as a marker of T cell activation. However, sustained high levels of PD-1 expression are associated with significantly decreased T cell effector functions [38]. This model allows us to probe how specific immune parameters of interest interact with other immune response variables, in order gain insight into how the parameter contributes to the overall immune environment of the host. Modeling results show increasing parasite burden is associated with increasing percent IL-10+ CD4+ T cells. In turn, higher frequencies of IL-10+ CD4+ T cells were led to higher levels of PD-1 + CD4+ T cells, and PD-1+ CD4+ T cells were associated with higher parasite burdens. This suggests a positive feedback loop between parasite burden, IL-10, and PD-1 pathways. Further, enhanced percent PD-1+ CD4+ T cells (at LeishVet score 1 and 3) and IL-10+ CD4+ T cells (at LeishVet score 1 and 2) were both associated with reduced proportions of protective IFN-$\gamma$ + CD4+ T cells.

Consistently, this model generated evidence suggesting tick-borne pathogen co-infections exacerbate CanL. Tick-borne pathogen co-infection status was strongly associated with an increased probability of disease progression from LeishVet score 1 to score 2, as well as from LeishVet score 2 to score 3. We observed that co-infected dogs were associated with increased anti-*Leishmania* antibody levels, percent IL-10+ CD4+ T cells, and percent PD-1+ CD4+ T cells. This enhanced regulatory CD4 T cell phenotype associated with co-infected dogs may negatively affect cellular immunity and host cell leishmanicidal activity. In agreement, receiving tick preventative treatment was strongly negatively associated with parasite load. The model results suggest tick preventative treatment leads to increased future percent IFN-$\gamma$ + CD4+ T cells which are known to be protective but, counter-intuitively, also led to increased future percent PD-1+ CD4+ T cells.

Canine CD8+ T cells are capable of influencing *Leishmania*-infected macrophage microbicidal activity [39], therefore we also included CD8+ T cell parameters in this model to see if any associations were revealed. We observed a strong negative effect of tick preventative treatment on future CD8+ T cell proliferation (posterior probability 0.928). This could imply CD8 + T cells are involved in tick-borne co-infections in dogs with underlying CanL. Another strong association we observed was between increased anti-*Leishmania* antibody levels and increased percent PD1+ CD8+ T cells throughout CanL progression (posterior probability 0.928, 0.944, and 0.717 at LeishVet scores 1, 2, and 3 respectively). High amounts of circulating antibodies during CanL can lead to kidney dysfunction and are usually observed during advanced clinical disease. Previous studies of human and experimental VL show CD8+ T cells display a high level of inhibitory receptors including PD-1 and a dysfunctional, exhausted phenotype during symptomatic disease [12, 40, 41].

The model identified outcomes associated with the risk of death due to progressive CanL disease. Unsurprisingly, pathogen load was associated with a higher risk of death. In terms of immune responses, we also found that increased frequencies of IL-10+ CD4+ T cells and IFN-$\gamma$+ CD4+ T cells both showed a strong association of increased risk of death due to progressive

CanL. This agrees with our findings that both regulatory and inflammatory CD4+ T cell functions are activated in dogs battling CanL. From the hazard ratios obtained for each association, we saw that increased percent IL-10+ CD4+ T cells had a much stronger effect on the likelihood of death, with an HR of $\sim 8.8$x, compared to percent IFN-$\gamma$+ CD4+ T cells (HR $\sim 2.9$x). This indicates that although both cytokines are increased in dogs with severe disease, the relative contribution of IL-10 is more significant when it comes to mortality outcome.

A benefit of this model framework is that it allows researchers to insert different immune parameters and measure their relationship to disease outcome and pathogen load to identify candidates for experimental testing. Furthermore, the model allows measuring the association of these longitudinal readouts with the risk of death due to severe cases of CanL. This model incorporates the percent of T cells expressing IFN-$\gamma$, IL-10, or PD-1 but does not capture the magnitude of that expression. Future model iterations could include additional continuous measurements such as concentration of secreted IFN-$\gamma$ or IL-10 measured by ELISA, or cytokine and PD-1 mean fluorescence intensities, which may further illuminate if relative levels of these inflammatory and regulatory proteins alter the probability outcomes observed by the model. Additionally, this model does not take into account other immune or stromal cell types expressing IFN-$\gamma$, IL-10, PD-1, or their respective receptors. Therefore, we can only draw conclusions about the role of these proteins with respect to their expression by the cell types included in this model. Some aspects of Leishmaniasis biology may be difficult to predict within a model framework. *Leishmania* parasites have evolved multiple immune evasion strategies that exist in a dynamic interaction with the host microbicidal mechanisms. Parasitized host cells undergo epigenetic and transcriptional changes to subvert the host cell polarization stage in favor of the parasite [42]. Finally, there are numerous other immune cell types, cytokines, and receptors involved in the pathogenesis of CanL that may interact directly or indirectly with the immune parameters included in this model, highlighting the complexity of this disease and the need for more inclusive datasets to improve model utility to predict immune responses and clinical outcomes. In addition, opportunities to explore causal direct and indirect processes through counterfactual formulations abound with this approach and may help to better understand the processes of disease progression in future studies.

Our main purpose was to better understand CanL disease progression over time, its relationship to different immunological responses, and its associations with the risk of death. Despite a large number of research studies on the immune response occurring during *Leishmania* infection, the entirety of the underlying mechanisms and their relationship to disease progression remain to be elucidated. The novel contribution of this model comes from the inclusion of immune responses as outcomes and possible drivers of disease progression, as well as the incorporation of different time-series and longitudinal model techniques into the within-host joint modeling framework of longitudinal and time-to-event data. We found important results regarding the effects of inflammatory and regulatory responses in disease progression. We also demonstrated a Bayesian model including immune parameters improved the predictive trajectories of pathogen load and disease classification, producing more reasonable and interpretable results compared to a model that does not take immune parameters into account [3]. This work can provide insight and new immune interactions associated with canine Leishmaniasis disease progression.

There are additional factors that can influence CanL dynamics that were not included in the present model but could represent future directions to include in models of this complex disease. For instance, individual host genetic predispositions such as expression of certain Major Histocompatibility Complex class II (Dog Leukocyte Antigen) alleles have been shown to be associated with increased susceptibility to visceral leishmaniasis and, although all dogs in this cohort were of the same hound breed, allele expression was not assessed and could reveal

interesting associations with disease progression within an expanded model [43]. *Leishmania* parasites also exhibit significant genetic diversity from host to host or even within a single host. *Leishmania* parasites undergo meiosis and sexual reproduction in the sand fly midgut, resulting in a unique polyclonal inoculum [44]. However, whole genome sequencing of parasites derived from vertically infected US dogs shows evidence that parasites within this enzootic cycle are more clonal [45]. That said, polymorphisms affecting virulence or tissue tropism may exist within the cohort due to genetic drift. It would be interesting to include parasite genomic data within a model to identify genes associated with developing severe disease. Further, this cohort regularly encounters local insect populations. Arthropod bites contain immunomodulatory salivary proteins which can influence skin immune microenvironments. Sand fly saliva has potent vasodilation activity, chemoattraction of monocytes and other myeloid cells, and contributes to an anti-inflammatory environment which may act on recruited phagocytes to be more permissive host cells [46]. Tick salivary proteins have anti-inflammatory effects on cytokine signaling, and cell proliferation, and inhibit complement cascades [47]. The humoral response to vector salivary proteins can be quantified via serology to estimate vector bite exposure levels. This seroreactivity could be included in future model iterations.

In this study, our primary emphasis was on utilizing posterior predictive trajectories as a principal tool to assess the adequacy of the Bayesian statistical model. While both posterior predictive plots and leave-one-out cross-validation (LOOCV) are conventional techniques in Bayesian data analysis for evaluating model fit, our decision to prioritize posterior predictive plots arises from the anticipated computational demands associated with implementing LOOCV within our modeling framework. Therefore, our future efforts are focused on optimizing the computational efficiency of our framework. Once we have successfully streamlined the computational processes and improved running times, we will turn our attention to incorporating LOOCV into our evaluation strategy. This staged approach ensures that we effectively address computational challenges before integrating LOOCV, maintaining a clear trajectory toward a comprehensive assessment of our proposed modeling methodology.

Based on the sensitivity analysis performed on the presented Bayesian model, it was found that the posterior density plots of the model parameters associated with pathogen load, antibody levels, and disease status were relatively similar across the different prior specifications considered. This suggests that the choice of prior had little impact on the resulting posterior inference for this model parameters. These results provide evidence of the robustness of the Bayesian inference to different prior specifications and support the validity of the chosen prior distributions, which also serves as evidence for the Bayesian model presented in our previous work [3]. The posterior density plots for the association parameters between the longitudinal and survival models were also relatively similar across the different prior specifications. However, we noticed that the selected prior specifications for parameters associated with the inflammatory and regulatory responses of the immune system were affected by this choice, suggesting further investigation. It is important to note that the sensitivity analysis was limited to a small number of prior specifications, and additional analyses may be needed to fully explore the impact of different prior choices on the model inference. Overall, these findings provide partial support for the reliability of the Bayesian analysis presented in the current work.

The generality of the framework employed here enables it to be used to model and predict disease outcomes of other infectious diseases as well as identify important signaling of increased or decreased risk of death due to severe cases of the disease. In applications where disease transmission between individuals is considered, it would be straightforward to adapt our model to define a linked model of the longitudinal within-host model of disease

progression with a transmission model (between-host) of infectious diseases. A full understanding of infectious diseases requires two distinct components: an understanding of transmission dynamics between hosts and an adequate model of within-host disease progression processes. These processes may be influenced by each other, particularly if time and other factors (internal and external) are considered. Nevertheless, simultaneously modeling at these disparate scales is practically and computationally challenging. We believe that, in future studies, this type of linked model approach may provide a greater opportunity for medical practitioners and researchers to work together, and contribute to the understanding of the drivers, treatments, and control of infectious diseases. Fortunately, the use of Bayesian techniques enables this flexibility in model specification, albeit at a potentially greater computational cost. This remains, however, a promising avenue for future research.

## Supporting information

**S1 Fig. PPT of pathogen load.**
(TIF)

**S2 Fig. PPT of antibody levels.**
(TIF)

**S3 Fig. Posterior marginal probabilities of disease status.**
(TIF)

**S4 Fig. PPT of proportion of IFN−$\gamma$+ of CD4+.**
(TIF)

**S5 Fig. PPT of proportion of proliferation of CD4+.**
(TIF)

**S6 Fig. PPT of proportion of proliferation of CD8+.**
(TIF)

**S7 Fig. PPT of proportion of IL−10+ of CD4+.**
(TIF)

**S8 Fig. PPT of proportion of PD−1+ of CD4+.**
(TIF)

**S9 Fig. PPT of proportion of PD−1+ of CD8+.**
(TIF)

**S1 File. Markov chain monte carlo summary and diagnostics.**
(PDF)

**S2 File. Sensitivity analysis.**
(PDF)

## Author Contributions

**Conceptualization:** Felix M. Pabon-Rodriguez, Grant D. Brown.

**Data curation:** Breanna M. Scorza.

**Formal analysis:** Felix M. Pabon-Rodriguez.

**Funding acquisition:** Breanna M. Scorza, Christine A. Petersen.

**Investigation:** Felix M. Pabon-Rodriguez, Grant D. Brown, Breanna M. Scorza, Christine A. Petersen.

**Methodology:** Felix M. Pabon-Rodriguez, Grant D. Brown.

**Project administration:** Grant D. Brown, Christine A. Petersen.

**Resources:** Christine A. Petersen.

**Software:** Felix M. Pabon-Rodriguez.

**Supervision:** Grant D. Brown, Christine A. Petersen.

**Validation:** Grant D. Brown, Christine A. Petersen.

**Visualization:** Felix M. Pabon-Rodriguez, Breanna M. Scorza.

**Writing – original draft:** Felix M. Pabon-Rodriguez.

**Writing – review & editing:** Felix M. Pabon-Rodriguez, Grant D. Brown, Breanna M. Scorza, Christine A. Petersen.

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
