## [Decision Letter · Decision Letter 0]

27 Oct 2023

PONE-D-23-30405Within-Host Bayesian Joint Modeling of Longitudinal and Time-to-Event Data of Leishmania InfectionPLOS ONE

Dear Dr. Pabon-Rodriguez,

Thank you for submitting your manuscript to PLOS ONE. After careful consideration, we feel that it has merit but does not fully meet PLOS ONE’s publication criteria as it currently stands. Therefore, we invite you to submit a revised version of the manuscript that addresses the points raised during the review process.

Please, carefully review the points raised by the reviewers and make the necessary revisions in the models. If you choose to keep specific elements and deviate from the suggestions made, we kindly request that you provide a comprehensive and well-founded justification for your choices.

We look forward to receiving your revised manuscript.

Kind regards,

Vinícius Silva Belo

Academic Editor

PLOS ONE

Journal Requirements:

Research reported in this article was supported by the National Institutes of Allergy and Infectious Diseases (NIAID) of the National Institutes of Health (NIH) of the United States of America under award number R01AI139267-03 and by MFHA18441000. This work was also performed while Breanna M. Scorza was supported by NIH/NIAID T32AI007260, which was the basis for data collection.

Research reported in this article was supported by the National Institutes of Allergy and 869

Infectious Diseases (NIAID) of the National Institutes of Health (NIH) of the United 870

States of America under award number R01AI139267-03 and by MFHA18441000. This 871

work was also performed while Breanna M. Scorza was supported by NIH/NIAID 872

T32AI007260, which was the basis for data collection. The content is solely the 873

responsibility of the authors and does not necessarily represent the official views of the 874

National Institutes of Health. Summary graphic figure created using BioRender.com. 

Research reported in this article was supported by the National Institutes of Allergy and Infectious Diseases (NIAID) of the National Institutes of Health (NIH) of the United States of America under award number R01AI139267-03 and by MFHA18441000. This work was also performed while Breanna M. Scorza was supported by NIH/NIAID T32AI007260, which was the basis for data collection.

Reviewers' comments:

Reviewer's Responses to Questions

**Comments to the Author**

1. Is the manuscript technically sound, and do the data support the conclusions?

Reviewer #1: Partly

Reviewer #2: Yes

Reviewer #3: Yes

2. Has the statistical analysis been performed appropriately and rigorously? 

Reviewer #1: I Don't Know

Reviewer #2: Yes

Reviewer #3: Yes

3. Have the authors made all data underlying the findings in their manuscript fully available?

Reviewer #1: Yes

Reviewer #2: Yes

Reviewer #3: Yes

4. Is the manuscript presented in an intelligible fashion and written in standard English?

Reviewer #1: Yes

Reviewer #2: Yes

Reviewer #3: Yes

5. Review Comments to the Author

Reviewer #1: The manuscript by Pabon-Rodriguez et al. proposes a joint longitudinal and temporal model to better assess the binomial dog_L. infantum infection, considering the interaction between three components of dog infection: parasite load, antibody level and disease severity. This is a statistical model, Bayesian modeling, focused only on the dog infection event. The authors hypothesized that using this approach, it is possible to infer the probability of occurrence of components attributed to canine Leishmania infection that are neither repeatable nor random, guiding epidemiological and clinical decisions. The approach taken by the authors is relevant and could provide new insight into the predictive course of binomial dog_L. infantum infection. However, as leishmaniasis are multifactorial disease the focus only on dog infection data is a fragility and could show an inadequate/incomplete discussion.

Questions to be address through the manuscript:

Canine leishmaniasis is a reservoir-parasite interaction considered a complex system, as it is multifactorial, unpredictable, and dynamic, forming a biological unit that can be in constant change, depending on changes in the environment (doi:10.1155/2018/3296893; doi:10.3390/vetsci9080387; doi:10.1186/1756-3305-4-86). Authors are encouraged to consider the frequency of susceptible reservoirs and vectors to the parasite, animals’ species that guarantee the circulation of L. infantum in nature or in peridomestic, within a specific time and space. This information is relevant due to L. infantum infections that cause VL have been described in several species of wild, synanthropic and domestic animals, frequency and diversity of sand flies related to transmission. Vector’s saliva is a potent pharmacologically active fluid that directly affects the haemostatic, inflammatory and immune responses of vertebrate hosts (doi:10.1111/j.1365-3083.2007.01964.x). In fact, a start for inflammatory and regulatory immune responses, even before dog infections by L. infantum.

Other elements of the system configure biological filters that impact the maintenance/survival of the parasite's genome, a "stronger phenotypic" parasite. Phenotypic heterogeneity in Leishmania spp is a factor that must be taken into consideration due to the probability of the frequency of a given phenotype influencing the infection course/outcomes (doi: 10.1186/s13071-015-0837-y; doi: 10.1186/s13071-020-3972-z; doi: 10.1016/j.protis.2017.11.004; doi: 10.1017/S003118202200097X). Therefore, when considering other biological elements in the Bayesian Joint Modeling the predictive results will be more robust.

Furthermore, it is important to keep in mind that intracellular parasites such as Leishmania spp. have evolved intricate strategies to subvert host cell functions for their own survival (doi:10.1016/j.pt.2021.09.009). The parasite can remain in the host without causing disease, as a biological advantage. These questions must be considered in the discussion of this article, as highlighting one of the factors not expected by used mathematical model.

What breeds and "Major Histocompatibility Complex" (MHC) of the dog cohort of this study? Dog MHC, which is known as "Dog Leukocyte Antigen" (DLA) has been studied and shown to be very variable. So, as this study contemplates cellular immune response approach is relevant to show the predominant MHC (highest frequency) from the selected dog cohort, its relationship with the endemic area and infection outcomes. Various DLA alleles and haplotypes have been shown to be associated with susceptibility to visceral leishmaniasis (doi: 10.1007/s00251-003-0545-1).

Reviewer #2: Dear authors,

The manuscript develop a model to determine the progression of leishmania infection and describe the mechanisms causing the disease progression. I think this is a relevant tool for clinical practice in veterinary medicine.

Congratulations to the authors, great work with relevant results

Reviewer #3: Line 203 "the state components are organized into subject-specific a row-vector", move "a" before subject-specific or delete "a".

Formula 1. why logit instead of log? I think lognormal is a more common model for Pathogen Load than logit. Logit is more common for proportion. I think the LOD for pathogen load was 10 then logit of 10 or any value larger than 1 can not be defined mathematically. The question of use of logit also applies to many other variable models defined below. A justification is necessary here. Also please remind readers Pi,t is Pathogen Load around the formula. The definition was way back.

Figure 2. "Graphical display of how the longitudinal outcomes are structurally associated with the risk of death." I do not see this structural association in this figure. The only thing related to death is the hazard plot, and it is not connected to anything else in the figure.

Line 310, "To characterize the censoring process, we treated all logit-scaled values below a corresponding transformed LOD∗ as left-censored and set them to be NA (“missing”)." I am confused here. What is this LOD∗ and how is it related to LOD in the text above? What is the value for LOD*? Is this a natural censoring by a limit of detection or artificial due to the logit transformation? It should be noted that censored data is not "missing", the term "missing" was confusing here. Most commonly a indicator is used for censoring together with the value censored.

Line 355 "To measure the impact of longitudinal outcomes on the risk of death due to progressive Leishmaniosis, a survival submodel is needed to examine the association between the two disease processes." What two disease processes? I thought the survival model is for death here.

Throughout the paper, I would think the time to event model was for disease before reading the paper carefully. And that model is actually for death, not for disease. I think it would be nice to make this clear earlier.

6. PLOS authors have the option to publish the peer review history of their article (what does this mean?). If published, this will include your full peer review and any attached files.

Reviewer #1: No

Reviewer #2: No

Reviewer #3: No

---

## [Author Response · Author response to Decision Letter 0]

29 Nov 2023

A PDF file has been submitted as an attachment, where we address specific reviewer and editor comments.

---

## [Decision Letter · Decision Letter 1]

2 Jan 2024

Within-Host Bayesian Joint Modeling of Longitudinal and Time-to-Event Data of Leishmania Infection

PONE-D-23-30405R1

Dear Dr. Pabon-Rodriguez,

We’re pleased to inform you that your manuscript has been judged scientifically suitable for publication and will be formally accepted for publication once it meets all outstanding technical requirements.

Kind regards,

Vinícius Silva Belo

Academic Editor

PLOS ONE

Additional Editor Comments (optional):

Congratulations!

Reviewers' comments:

Reviewer's Responses to Questions

**Comments to the Author**

1. If the authors have adequately addressed your comments raised in a previous round of review and you feel that this manuscript is now acceptable for publication, you may indicate that here to bypass the “Comments to the Author” section, enter your conflict of interest statement in the “Confidential to Editor” section, and submit your "Accept" recommendation.

Reviewer #3: All comments have been addressed

2. Is the manuscript technically sound, and do the data support the conclusions?

Reviewer #3: Yes

3. Has the statistical analysis been performed appropriately and rigorously? 

Reviewer #3: Yes

4. Have the authors made all data underlying the findings in their manuscript fully available?

Reviewer #3: Yes

5. Is the manuscript presented in an intelligible fashion and written in standard English?

Reviewer #3: Yes

6. Review Comments to the Author

Reviewer #3: (No Response)

7. PLOS authors have the option to publish the peer review history of their article (what does this mean?). If published, this will include your full peer review and any attached files.

Reviewer #3: No

---

## [Editor Report · Acceptance letter]

30 Jan 2024

PONE-D-23-30405R1 

PLOS ONE

Dear Dr. Pabon-Rodriguez, 

I'm pleased to inform you that your manuscript has been deemed suitable for publication in PLOS ONE. Congratulations! Your manuscript is now being handed over to our production team.

Kind regards, 

on behalf of

Dr. Vinícius Silva Belo 

Academic Editor

PLOS ONE